# A Novel Influence Analysis-Based University Major Similarity Study

**Ningqi Zhang [1], Qingyun Li [2,\*], Sissi Xiaoxiao Wu [1], Junjie Zhu [1] and Jie Han [3]**

1    College of Electronics and Information Engineering, Shenzhen University, Shenzhen 518060, China; 2070436026@email.szu.edu.cn (N.Z.); xxwu.eesissi@szu.edu.cn (S.X.W.); 2110436015@email.szu.edu.cn (J.Z.)
2    Teaching and Learning Centre, Lingnan University, Hong Kong 000852, China
3    School of Science and Technology, Hong Kong Metropolitan University, Hong Kong 000852, China; chan@hkmu.edu.hk
\*    Correspondence: rainli@ln.edu.hk

**Abstract:** In the field of education, investigating the relationships between different majors in universities is an important topic in current educational research. The application of social networks from informatics provides new opportunities and potentials for the field of education. Due to the complexity of social interactions, the social network connections surrounding individuals exert a significant influence on their daily decision-making processes. This paper aims to introduce the social network and influence analysis theories from informatics into the field of education, regarding major as a variable, and comparing and analyzing the influence relationships between majors. An empirical study was conducted, involving the collection of questionnaire data on graduates' evaluations of various aspects of their university experiences across different majors. The evolution of this model follows the DeGroot opinion dynamics with the inclusion of stubborn nodes. By defining leader majors and general majors based on the data and modeling the questionnaire data as the outcome of a discrete random process, an influence matrix is ultimately generated through the opinion dynamic model. Through this modeling approach, we revealed the underlying influence relationships between different disciplines (majors). These findings provide schools with insights to adjust the directions of discipline cultivation, and offer new perspectives and methods for the study of majors in higher education.

**Keywords:** higher education; major similarity; data mining; influence matrix; influence analysis; opinion dynamics model; social network

## 1. Introduction

As students begin their journey towards higher education, choosing a college major is one of the most critical decisions that students have to make during their academic journey. It not only shapes their careers but also the quality of life they lead in the future. However, different university majors come with different challenges, skills, and opportunities. Sometimes students may make the wrong choices, leading to dissatisfaction and uncertainty after graduation [1]. These wrong choices may stem from their inadequate understanding of those majors, as there are clear differences and similarities in subject knowledge, skill development, and employment direction among disciplines and majors [2]. Therefore, analysis and research on the connections between different majors is of significant importance in promoting students' comprehensive development and enhancing the quality of education [3].

A major similarity study that compares and contrasts several majors may help students narrow down their choices, choose a major that is the best fit for them, and adapt to the requirements of different disciplines (majors). In addition, educational institutions could also optimize their teaching strategies and curriculum design more effectively [4]. Moreover, as the economy changes and new professions emerge, certain majors may become more

lucrative than others. A major similarity study can examine how majors adapt to these changes, such as adding new courses or emphasizing certain skills [5,6].

Traditionally, a university major similarity study would compare the required courses, career paths, and overall curriculum of two or more majors, and thus provide insights into the majors' similarities from the alumni's viewpoint to help prospective students make informed decisions about their major choices. However, such traditional educational research mainly relies on methods such as questionnaire surveys and statistical analysis [7,8]. In practice, the questionnaire survey method usually requires the involvement of domain experts to design appropriate questions, and the general statistical analysis usually falls short when conducting an influence analysis between different majors. Therefore, we hope to apply informatics technology to the field of education and provide new insights for professional similarity research from a new perspective.

This work aims to employ data mining technology for modeling and analyzing educational influence using school datasets. Data mining, as an interdisciplinary field, serves the purpose of extracting valuable information from vast amounts of data and unraveling underlying patterns that aid decision makers in adjusting market strategies, mitigating risks, and making informed decisions. Its applications span across diverse domains such as data statistics, market analysis, and production management. In the context of education, specific scenarios exist in which data mining finds relevance and practical application [9–13]. In [13], the author proposed a combined approach of social network analysis and educational data mining, which was used to study the impact of communication networks, behavior networks, and the combination of these two networks on students' academic performance. This kind of related research makes people aware of the importance of social networks.

A social network refers to a collection of points (social actors) and edges between points (relationships between actors). The analysis of a social network focuses on the relationships between the social actors; the patterns of which would affect the actors' actions [14]. Estimating the influence matrix of a social network is very challenging research. In the domain of social network analysis, matrices offer a viable approach for representing the intricate structures of social networks [15]. Within these matrices, the elements serve to signify the connections or ties that exist between actors. Graphical depictions of networks can incorporate weighted edges, with the elements within matrices assuming values that reflect the strength of the relationships between actors. Previous research endeavors have predominantly focused on extracting the node reputations within the network. The edges of the network can be effectively expressed as the relationships between nodes, encompassing various forms such as "agreement", "voting", and "recommendation" [16]. In this work, we will apply an informatics social network technique to the school questionnaire data to analyze the underlying influence relationships between different majors. Specifically, our survey responses have been gathered from graduates of a liberal arts University in Hong Kong. The objective of this survey is to capture alumni's opinions regarding the quality of courses and the learning environment offered by their alma mater. Drawing upon this data, the majors offered at the University are considered as nodes within a social network. By defining leader majors and general majors based on the data and modeling the questionnaire data as the outcome of a discrete random process, an influence matrix is ultimately generated through the opinion dynamic model. The focus of our research is to explore and analyze the profound interconnections between majors from a novel perspective by mining this data set.

## 2. Overview of Data Model

In 1974, DeGroot proposed a viewpoint dynamics model that explains how team members converge their opinions and adjust their own opinion distributions to reach consensus after gaining knowledge of the subjective opinions of other members [17]. It is postulated that a latent dynamic process precedes the completion of the questionnaire by the graduates. Within this process, alterations in the opinions of certain majors can impact the opinions of corresponding majors reflected in the questionnaire, thus exerting an influence

on the opinions of other majors. This is consistent with the idea of the DeGroot model. According to the DeGroot model, interactions between users would ultimately make the whole group tend to be consistent. Some scholars proposed a simple but insightful opinion dynamics model based on the DeGroot model [18,19], which examined the traditional first-order opinion consensus algorithm with a static symbolic interaction graph.

Our previous work [20] proposed an opinion dynamic model which introduced the concept of opinion leaders into the DeGroot model. Wu et al. found that, even with the inclusion of opinion leaders, the process of opinion convergence still occurs. However, the introduction of opinion leaders leads to a divergence in the opinions of the group on specific topics, rather than achieving consensus among the nodes. It has been proven that, if we can reach the steady-state opinion distribution of nodes, the model could accurately mine the influence between nodes. In our work, we would integrate the opinions of students in the same major to obtain the opinion distribution of this major through the tensor dimension reduction. In other words, by using tensor dimension reduction, each major could be regarded as a vector. Some majors will be regarded as leader majors, also called stubborn nodes in the social graph, whose opinions cannot be swayed by other nodes [20–22]. Due to historical reasons or their own characteristics, the leader majors have a huge impact on other majors and will dominate the DeGroot public opinion dynamic model, as they are often the ones who influence the opinions of others and are rarely affected by other majors. By employing tensor dimension reduction and defining opinion leaders, the opinion distributions of each major would be extracted from the questionnaire data, and the final influence matrix among different disciplines (majors) would be generated through the opinion dynamics model.

### 2.1. Tensor

In computer science, it is essential to store such data in appropriate structures. For instance, images can be treated as two-dimensional arrays composed of pixels, with each pixel represented by a triple which denotes the RGB values. For instance, an image can be transformed into a higher-order array, as depicted in Figure 1. While scalars, vectors, and matrices can be considered as special cases of tensors, tensors are primarily used to store high-order arrays [23]. Figure 2 illustrates that scalars are zero-order tensors, and vectors are first-order tensors.

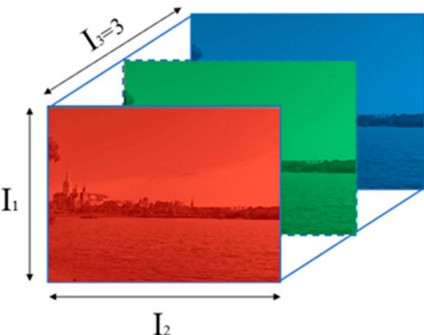

**Figure 1.** Each pixel could be considered as a triple composed of RGB.

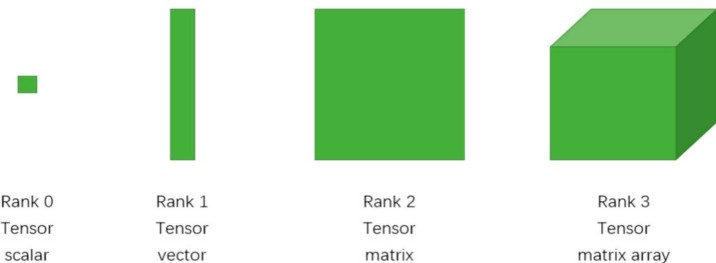

**Figure 2.** A tensor is an N-dimensional array of data.

In reality, a substantial portion of the data we encounter consist of high-dimensional tensors. For instance, videos encompass temporal, visual, and auditory information. Images can be regarded as three-dimensional tensors, resulting in videos being represented as five-dimensional tensors. Processing high-dimensional tensors is more complex compared to low-dimensional tensors due to the involvement of time and increased computational requirements. To address this, we employ dimensionality reduction techniques by mapping certain dimensions onto others [24], thereby reducing the tensor's overall dimensions. One simple approach for dimensionality reduction is multiplying a tensor with a vector, resulting in a lower-dimensional tensor, as shown in Figure 3.

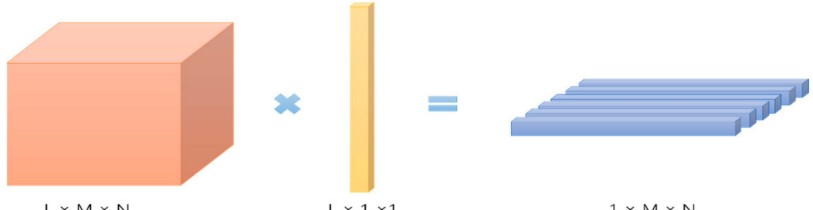

**Figure 3.** Multiply tensor and vector to reduce the dimension.

There are various methods which can reduce the dimensions of a tensor and other ways to save the information from data. To reduce the storage burden, [25] proposes a novel use of the row-product random matrices in random projection, which is called Tensor Random Projection (TRP), formed as the Khatri–Rao product of a list of smaller dimension-reduction maps. In [26], the author proposed a dimension-adaptive quadrature method to reduce the dimensions of tensor automatically. In [27], the author uses nonnegative Tucker decomposition (NTD), which obtains a set of smaller core tensors by finding a set of common projection matrices of tensor objects, and finally accomplishes the dimension reduction of tensors.

### 2.2. Opinion Dynamic Model

By using the method introduced in the previous section, the opinion vector of each key point can be obtained based on the answers to the questionnaire. Subsequently, we can introduce the DeGroot opinion dynamic model by incorporating stubborn nodes to infer the influence matrix based on node opinions.

The opinion dynamic model assumes that, within a group, members possess an initial opinion pertaining to a specific topic, referred to as the "initial state". Upon assimilating the opinions of other group members, a weighted aggregation approach is employed to incorporate diverse perspectives, resulting in the adjustment of one's own opinion on the topic. This process can be conceptualized a form of opinion integration. Eventually, each member's opinion converges to a state of equilibrium, known as the "steady state". This process from the "initial state" to the "steady state" represents how group members gather the opinions of others, alter their opinion distributions, and ultimately attain consensus by learning from the subjective opinions of fellow members. In the model proposed by Wu et al., after adding opinion leaders to the DeGroot model, the opinion dynamic process can still converge and the steady states will be decided by the opinion leaders. We will utilize this property to analyze how the leader majors affect other majors in the university system.

Within the university context, it is evident that different majors have different opinions on the same questionnaire. Moreover, it is important to recognize that these opinions are not entirely independent but rather influenced by majors that are closely aligned with their respective fields of study. Consequently, we assume that there is an underlying interactive process before each major fills out their questionnaire. In this process, each major will be influenced by some of the other majors to change their initial opinions, and finally form their own opinions which are then presented in the questionnaire.

We therefore consider the majors in the university as nodes in the network. We define the opinion matrix of $N$ nodes at time t as $X \in \mathbb{R}^{N \times K}$, whose K represents the dimension of

the majors' opinion vector on a specific topic in the questionnaire. In our model there are two kinds of nodes, i.e., there are $N_s$ stubborn nodes (corresponding to the leader majors) and $N_n$ non-stubborn nodes (corresponding to the normal majors). Then, the opinion matrix $X$ can be expressed as $Z \in \mathbb{R}^{N_s \times K}$ and $Y \in \mathbb{R}^{N_n \times K}$, respectively.

$$X = \begin{bmatrix} Z \\ Y \end{bmatrix}, \tag{1}$$

where $Z \in \mathbb{R}^{N_s \times K}$ and $Y \in \mathbb{R}^{N_n \times K}$ represents the opinions of K parameters for stubborn nodes and non-stubborn nodes. Since the stubborn nodes are not affected by the other nodes in the network, we define the influence matrix among these majors as $W \in \mathbb{R}^{N \times N}$.

$$W = \begin{bmatrix} I_{N_s} & 0 \\ B & D \end{bmatrix}, \tag{2}$$

where $W$ is a stochastic matrix, i.e., $\sum_j W_{ij} = 1$ for every $i$; $B \in \mathbb{R}^{N_n \times N_s}$ represents the influence of stubborn nodes on non-stubborn nodes, and $D \in \mathbb{R}^{N_s \times N_s}$ represents the mutual influence among the non-stubborn nodes. After that, our primary task is to calculate matrix $B$ and matrix $D$ by observing the nodes' opinions at their steady state.

The process of opinion diffusion among N members within a group on K topics at each time point in the discussion stage can be expressed as:

$$X_t = WX_{t-1}, \ \ t = 1, 2, \ldots \tag{3}$$

Herein, we made the same assumption as that made in [20,28,29]: that the network corresponding to $W$ is connected. Then, after the recursion, there will eventually be a steady state $X_\infty$. In this way, the underlying interactive process can be written as:

$$X_\infty = \lim_{t \to \infty} X_t = \lim_{t \to \infty} W^t X_0 \tag{4}$$

According to the division matrix multiplication principle, we can obtain:

$$\lim_{t \to \infty} W^t = \begin{bmatrix} I_{N_s} & 0 \\ (I-D)^{-1}B & 0 \end{bmatrix}. \tag{5}$$

As the opinions of the stubborn nodes in the network will not change throughout the whole process, i.e., $\lim_{t \to \infty} Z^t = Z^0$, we then substitute Equation (5) into Equation (2) to obtain:

$$\lim_{t \to \infty} Y^t = (I-D)^{-1} \cdot B \cdot \lim_{t \to \infty} Z^t. \tag{6}$$

Then, we replace the $\lim_{t \to \infty} Y^t$ with $Y$ and the $\lim_{t \to \infty} Z^t$ with $Z$ to obtain:

$$Y = (I-D)^{-1} \cdot B \cdot Z. \tag{7}$$

In order to solve $B$ and $D$, we need to construct a linear least-square fitting problem with regularization terms. The goal is to minimize the objective function:

$$\begin{aligned} \min_{B \geq 0, \ D \geq 0} & \rho \| (I-D)Y - BZ \|_F^2 + \| [B, D] \|_1 \\ \text{s.t } & (B+D)1 = 1 \\ & diag(D) = \updownarrow \end{aligned} \tag{8}$$

where $\rho$ is a parameter, which can be used to adjust the punishment of an L1 regular term $\| [B \ D] \|_1$ to prevent over fitting, $\updownarrow$ is a self-trust prior of the normal nodes, which represents the degree to which a node is susceptible to influence. A higher confidence indicates a lesser susceptibility to the influence of other nodes. When $\rho$ is smaller, the

sparsity of the solution will be higher, which makes the network structure in the model sparse and adapt to the real social network scene. The operator $diag(\cdot)$ represents taking out the diagonal elements of the matrix and forming a vector. We can solve Problem (8) by using the CVX toolbox [30].

## 3. Material and Methods

This study utilizes the secondary dataset derived from the questionnaire (see Appendix A) administered and provided by the University over a span of ten project cycles between 2002 and 2020. The survey, conducted biennially, aims to assess the perspectives of alumni regarding the quality of programs and the learning environment at the University. The data collection process involved the utilization of online platforms and postal mail questionnaires. The target respondents for the survey were recent graduates of the University, with a focus on those who had graduated within the preceding five years. Key areas of investigation included in the questionnaire are:

1. Level of importance of different skills and competencies obtained at the University for the alumni in the working environment.
2. Level of satisfaction with the education the University provided in terms of nurturing different skills and competencies of students.
3. Alumni's learning and living experiences at school, and views on supporting staff.
4. Alumni's anonymous job information and their engagement with the University after graduation.

A diverse range of over 30 majors actively engaged with and participated in this survey. The University, in its pursuit to comprehend the perspectives of its recent alumni regarding the quality of its programs and the learning environment, diligently conducts a biennial survey. The invaluable data collected through this survey will play a pivotal role in facilitating the university's ongoing endeavors to enhance the design of its programs. By leveraging this information, the University aims to equip future students with the essential skills and knowledge required to adeptly tackle the multifaceted challenges arising from the dynamic and ever-evolving demands of the twenty-first century.

This section can be divided by subheadings. It should provide a concise and precise description of the experimental result and their interpretation, as well as the experimental conclusions that can be drawn.

### 3.1. Data Preprocessing

Due to the dynamic nature of the questionnaire, which involves the addition and deletion of questions in each iteration, the dataset collected exhibits inherent differences. Consequently, it is imperative to preprocess the collected data before direct utilization due to the presence of numerous missing values and extraneous information. Questions that lack over 15% of responses were excluded from further analysis.

Moreover, considering the research objective of investigating potential interdependencies among majors through questionnaire completion, specific questions related to demographic information, such as "company size", "current salary" and "industry", were deemed irrelevant, as they do not involve any deliberation process prior to questionnaire completion. Additionally, questions that lack variation in response options were removed as imputing missing values and generating meaningful matrix values becomes challenging in such scenarios.

Following the aforementioned steps, a set of 53 questions (highlighted in yellow in Appendix A) with few missing values was selected as the pertinent information. Subsequently, to ensure the accuracy of results, graduate data containing more than 20% missing responses were excluded. However, even after these measures, the dataset may still have contained certain missing or invalid values, which were subsequently filled using the mode as it represented the most common selection among users for such issues. Furthermore, graduates who failed to provide their student numbers or filled "Others" in the "Major" option were excluded from the study, and we retained only the samples from the specific

29 options for majors provided on the questionnaire. Finally, we obtained 6090 samples from a total of 6771 questionnaire responses. The graduate data corresponding to each major were integrated separately based on the major number, resulting in the creation of 29 distinct matrices of graduate data.

### 3.2. Selection of the Leader Major

In the process of selecting leaders in the group, we believe that influential members should be more capable, have more resources, or be able to lead more members. Hence, we can identify leaders and normal members by evaluating their leadership skills or the number of resources that he possesses. Considering the characteristics of the data set, this experiment chose the latter method. That is, majors with a larger number of students are designated as "leadership majors" as they are more cohesive, have firmer opinions, and are less susceptible to external influences from other majors. Conversely, majors with smaller graduate populations are deemed as general majors, as they are more prone to being influenced by the opinions of other majors. We had a total of 29 majors. We therefore allocated the leader majors and the normal majors according to the ratio of 1:3 (this is also a proper ratio for which Problem (8) can be successfully solved [22]). Therefore, we chose the 7 majors with the largest number of students, namely Chinese, Cultural Studies, Translation, Accounting, Human Resource Management, Marketing, and Contemporary Social Issues and Policy, as our leader majors, and the other majors were classified as the normal majors.

### 3.3. Extraction of Majors' Opinion Vector

The data can be viewed as a tensor, with the three dimensions representing major, graduates, and questionnaire answers, respectively. In order to reduce the complexity of the model, the dimension of the tensor data needs to be reduced first. In Section 2, a simple method of tensor dimension reduction is introduced. The dimension of the data could be reduced by multiplying the tensor and the vector. Since the data do not contain any individual-specific information about graduates, each graduate was treated as an equal entity. Therefore, the length of our vector is L, and each element is $1/L$, where L is the number of students corresponding to each major. This approach enables us to accord equal weight to the opinions of students within the same major. Ultimately, the opinions of students in the same major could be integrated to obtain the opinion distribution of this major through the tensor dimension reduction. The opinion matrix, composed of majors and questionnaire questions, can be obtained by splicing the opinion vectors of each major obtained in the previous section. The vertical axis represents majors, and the horizontal axis represents their responses to various questions. In order to mitigate overfitting and obtain sparser solutions that are more suitable for real-world scenarios, the parameter $\rho$ is set to 0.9 and the confidence parameter $\updownarrow$ of normal nodes is set to 0.25 to achieve a good training performance. To solve this optimization problem, we can directly utilize the well-established numerical computing software MATLAB (version 2.0) along with the corresponding optimization toolbox, CVX [30]. We could ultimately obtain the optimal solution and output the figure of influence matrix by the function "imagesc" in MATLAB. The "imagesc" function converts the values in the matrix to different colors and paints them at the corresponding positions on the coordinate axis. The brighter positions denote a larger value in the influence matrix we plotted.

## 4. Results

Figure 4 illustrates the influence matrix derived using the aforementioned methodology. The index range of one to seven corresponds to leading majors, while the index range of eight to twenty-nine represents general majors. Based on the color bar on the right and the darkness of the highlighted part in the figure, we can observe the degree of influence between majors. Generally speaking, if the research content of two majors is more relevant,

the influence between them will be greater. A comprehensive analysis of our experimental results also further demonstrates this conclusion.

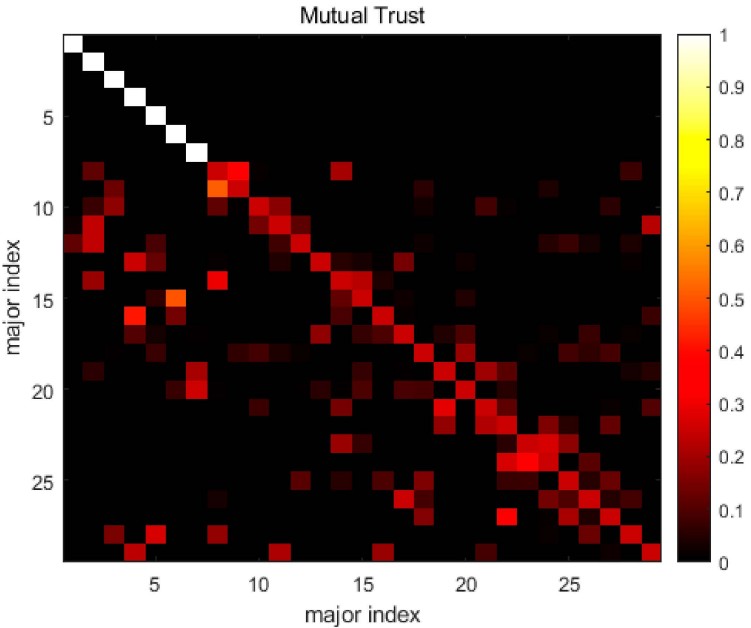

**Figure 4.** The influence matrix among majors.

Based on Figure 4, it can be deduced and analyzed that considerable reciprocal influence exists among majors that share similarities. The findings indicate the presence of mutual influence between Contemporary English Studies and Contemporary English and Education as shown in Figure 5. Specifically, the impact of Contemporary English Studies on Contemporary English and Education appears to be more pronounced. There clear similarities exist between these two majors from the perspective of disciplinary characteristic and objective, which may contribute to the substantial influence relationship observed between them.

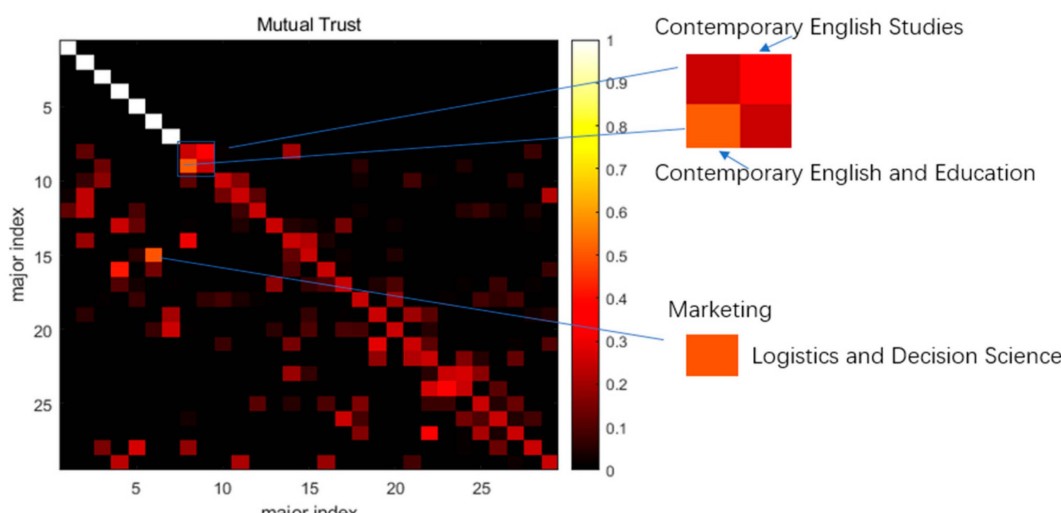

**Figure 5.** The first example of the influence matrix: the mutual influence among similar majors.

Furthermore, Marketing evidently exerts influence on Logistics and Decision Science. Marketing places emphasis on areas such as consumer behavior, market research, and market positioning, which align with the decision science and analytical techniques employed in Logistics and Decision Science. This alignment facilitates the formulation of

effective marketing strategies and decisions. Hence, the noteworthy impact of Marketing on Logistics and Decision Science is not surprising. In general, these majors demonstrate a significant level of similarity with one another.

The marked influence of Accounting on Information Systems can be observed in Figure 6. It is widely acknowledged that graduates majoring in accounting frequently encounter information systems in their professional endeavors, as they are adept at managing and auditing complex systems as part of their responsibilities in the workplace. Conversely, graduates majoring in Information Systems often engage in the establishment of comprehensive information systems and the management of databases as integral components of their jobs' requirements. As such, the interconnectedness between these two majors is to be expected.

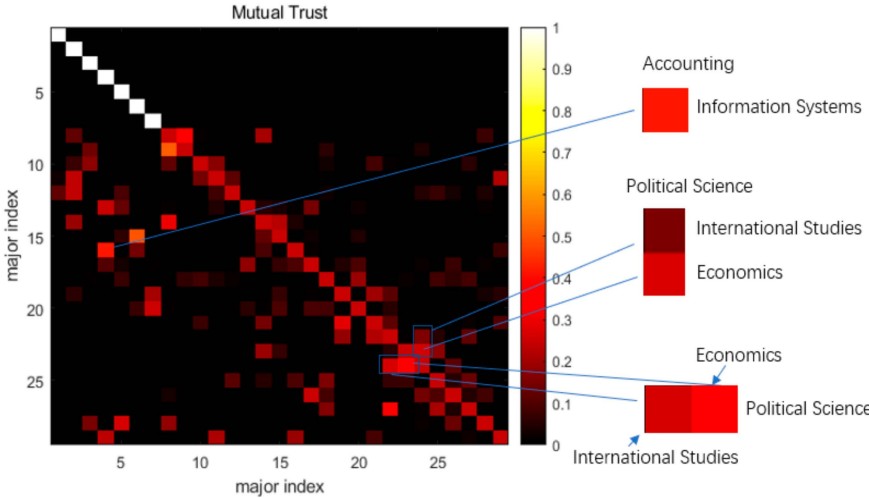

**Figure 6.** The second example of the influence matrix: the mutual influence among related majors.

Additionally, the results indicate that both International Studies and Economics have an influence on Political Science. It is universally acknowledged that economics and politics are inseparable, and the study of international relations cannot be separated from the support of political science. This logical connection is reflected in the influence exerted by these two majors. However, the influence of Political Science on Economics and International Studies is relatively less pronounced in Figure 6, yet they all belong to School of Social Science. Based on this result, universities can consider strengthening the connections between these three majors in various aspects.

To visualize the influence relationships among the majors at the University, we employed the Fruchterman–Reingold algorithm, available in the Gephi drawing software [31]. Gephi 0.10.1 is a piece of software used for visualizing and exploring all types of graphs and networks. Figure 7 illustrates the resultant visualization by Gephi. In this representation, the outer nodes correspond to leader majors, while the inner nodes represent general majors. The edges connecting the nodes are depicted as weighted arrows, with thicker edges indicating larger weights. Upon examining the visualization in a clockwise manner, it becomes apparent that the majority of leader majors exhibit thicker arrows pointing towards general majors. Conversely, a few leader majors display thinner arrows indicating influence towards general majors.

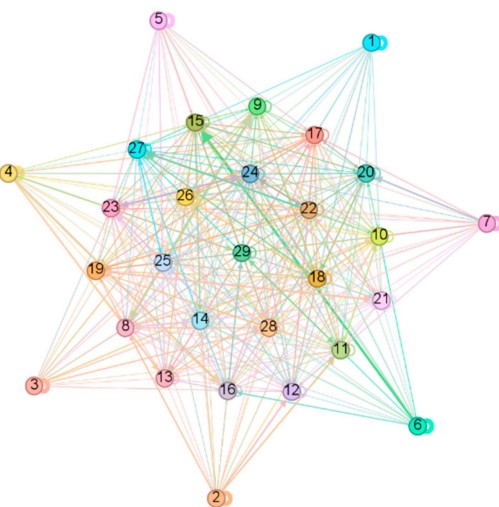

**Figure 7.** Visualization of majors' influence relationships with Gephi.

## 5. Conclusions

In conclusion, the primary objective of this study was to implement an influence analysis and uncover the underlying influence relationships between majors within questionnaire data from the perspective of alumni. To achieve this, we proposed an informatics data mining approach that leverages an opinion dynamic model to explore the mutual influences among various majors, utilizing data from a questionnaire completed by graduates. Through the examination of the generated model, our findings align with widely held opinions regarding the practical relevance of graduates' majors to their respective careers.

In addition, as we have mined the underlying influence between any two majors from our social network model, we can discover and analyze which majors within the university are closely connected and which lack interdisciplinary links. This information can help students make better choices when selecting majors, and facilitate academic exchanges with other majors. In addition, it also provides a good reference for optimizing teaching strategies and curriculum design, promoting communication between disciplines (majors), and promoting interdisciplinary cooperation.

Moreover, the proposed methodology proposed in this paper opens up possibilities to further mine data from graduates, providing researchers with a new perspective to further explore the differences and connections between majors and devise career development strategies tailored to different majors within educational institutions.

**Author Contributions:** Conceptualization, N.Z. and S.X.W. methodology, S.X.W. and Q.L.; software, N.Z.; validation, S.X.W. and Q.L.; formal analysis, S.X.W.; investigation, Q.L.; resources; data curation, Q.L.; writing—original draft preparation, Q.L.; writing—review and editing, J.Z., Q.L. and J.H.; visualization, Q.L.; supervision, S.X.W.; project administration, Q.L.; funding acquisition, Q.L. All authors have read and agreed to the published version of the manuscript.

**Funding:** No funding was received to conduct this study.

**Institutional Review Board Statement:** This survey is a standard institutional survey, started before 1995. Ethics Committee or Institutional Review Board approval not be required for the manuscript. The aggregate data are available in the University annual report: https://www.ln.edu.hk/media/publications-and-multimedia/university-publications/annual-report (1 March 2024).

**Informed Consent Statement:** Not applicable.

**Data Availability Statement:** Data available on request due to restrictions.

**Conflicts of Interest:** The authors declare no conflicts of interest.

# Appendix A  The Questionnaire from the Alumni Survey

**Survey of Alumni**
校友問卷調查

This survey aims to solicit alumni feedback regarding your level of satisfaction with the knowledge, skills and attitudes gained during your studies and experiences at the University. The information collected will help the University to improve the design of the Programmes and better equip future students to meet the challenges of the twenty-first century.

此問卷旨在收集大學校友在大學求學時期的各方面意見，包括所獲得的知識、技能、以及待人處事的態度等。所得資料，將幫助大學更完善設計課程，更有效培育未來學生，以迎接二十一世紀的挑戰。

**I.   EDUCATION PROVIDED BY THE UNIVERSITY 大學提供的教育**

There are two columns to each question. **Please indicate your response to each question by circling the appropriate score.**
以下各問題均分為左右兩欄，請於各欄圈出適當的分數。

**Column in the left:**
**LEVEL OF IMPORTANCE**
左欄：各項目的重要程度

**Column in the right:**
**SATISFACTION LEVEL OF THE EDUCATION ON YOU**
右欄：您對大學提供的各項教育的滿意程度

Please rate the importance of the following attributes for a university graduate to possess and demonstrate in a working environment.
請在左欄圈出適當分數，以表示您認為所列的各種知識、技能或個人特質在工作上的重要程度。

Please indicate to what extent you are satisfied with the education provided by the University regarding the following attributes.
請在右欄圈出適當分數，以表示您對大學提供的各項知識、技能或個人特質方面的培育的滿意程度。

1= Not important      5 = Very important
1=極不重要         5=極重要

1= Very dissatisfied      5 = Very satisfied
1= 極不滿意         5 = 極滿意
N = Unable to rate
N =不適用

| Level of importance to your work 有關能力在工作上的重要性 Not important → Very important 極不重要 → 極重要 | | | | | Skills/Competencies 技能或能力 | Level of satisfaction with the education 大學在這方面對您的培育的滿意程度 Very dissatisfied → Very satisfied 極不滿意 → 極滿意 | | | | | Unable to rate 不適用 |
|---|---|---|---|---|---|---|---|---|---|---|---|
| **A: LANGUAGE PROFICIENCY 語文水平** | | | | | | | | | | | |
| 1 | 2 | 3 | 4 | 5 | 1 Written Chinese 中文書寫 | 1 | 2 | 3 | 4 | 5 | N |
| 1 | 2 | 3 | 4 | 5 | 2 Putonghua 普通話 | 1 | 2 | 3 | 4 | 5 | N |
| 1 | 2 | 3 | 4 | 5 | 3 Written English 英文書寫 | 1 | 2 | 3 | 4 | 5 | N |
| 1 | 2 | 3 | 4 | 5 | 4 Spoken English 英語會話 | 1 | 2 | 3 | 4 | 5 | N |

| Level of importance to your work 有關能力在工作上的重要性 Not important → Very important 極不重要 → 極重要 | | | | | Skills/Competencies 技能或能力 | Level of satisfaction with the education 大學在這方面對您的培育的滿意程度 Very dissatisfied → Very satisfied 極不滿意 → 極滿意 | | | | | Unable to rate 不適用 |
|---|---|---|---|---|---|---|---|---|---|---|---|
| **B: NUMERICAL COMPETENCY & COMPUTER LITERACY 分析數據能力及電腦知識水平** | | | | | | | | | | | |
| 1 | 2 | 3 | 4 | 5 | 1 Data analysis ability 數據分析能力 | 1 | 2 | 3 | 4 | 5 | N |
| 1 | 2 | 3 | 4 | 5 | 2 Use of software, e.g. word processing, spreadsheet, database etc. 善用電腦軟件，如文字處理、簡報、數據庫等 | 1 | 2 | 3 | 4 | 5 | N |
| **C: ANALYTICAL & PROBLEM-SOLVING ABILITIES 分析及解決問題的能力** | | | | | | | | | | | |
| 1 | 2 | 3 | 4 | 5 | 1 Ability to foresee problems and plan 洞悉問題及規劃的能力 | 1 | 2 | 3 | 4 | 5 | N |
| 1 | 2 | 3 | 4 | 5 | 2 Ability to analyze and solve problems 分析及解決問題能力 | 1 | 2 | 3 | 4 | 5 | N |
| 1 | 2 | 3 | 4 | 5 | 3 Ability to articulate new ideas 表達新意念的能力 | 1 | 2 | 3 | 4 | 5 | N |
| 1 | 2 | 3 | 4 | 5 | 4 Ability to apply a systematic/logical approach to problem solving 透過有系統／邏輯方法解決問題的能力 | 1 | 2 | 3 | 4 | 5 | N |
| | | | | | 5. Creative and critical thinking 創意及批判性思考 | 1 | 2 | 3 | 4 | 5 | |
| **D: INTER-PERSONAL SKILLS 人際關係技巧** | | | | | | | | | | | |
| 1 | 2 | 3 | 4 | 5 | 1 Effective communication 能有效與人溝通 | 1 | 2 | 3 | 4 | 5 | N |
| 1 | 2 | 3 | 4 | 5 | 2 Ability to build rapport with people 能建立和諧的人際關係 | 1 | 2 | 3 | 4 | 5 | N |
| 1 | 2 | 3 | 4 | 5 | 3 Cooperation with colleagues 能與同事合作無間 | 1 | 2 | 3 | 4 | 5 | N |
| **E: MANAGEMENT SKILLS 管理技巧** | | | | | | | | | | | |
| 1 | 2 | 3 | 4 | 5 | 1 Time management 時間管理 | 1 | 2 | 3 | 4 | 5 | N |
| 1 | 2 | 3 | 4 | 5 | 2 Leadership 領導才能 | 1 | 2 | 3 | 4 | 5 | N |
| 1 | 2 | 3 | 4 | 5 | 3 Organization abilities 組織能力 | 1 | 2 | 3 | 4 | 5 | N |

| Level of importance to your work 有關能力在工作上的重要性 | | | | | Skills/Competencies 技能或能力 | Level of satisfaction with the education 大學在這方面對您的培育的滿意程度 | | | | | Unable to rate 不使用 |
|---|---|---|---|---|---|---|---|---|---|---|---|
| Not important → Very important 極不重要 → 極重要 | | | | | | Very dissatisfied → Very satisfied 極不滿意 → 極滿意 | | | | | Unable to rate 不使用 |
| colspan="6" | **F: INTERNATIONAL PERSPECTIVES 國際視野** | | | | | | |
| 1 | 2 | 3 | 4 | 5 | 1.Knowledge and understanding of current international affairs 能緊貼國際時事脈搏 | 1 | 2 | 3 | 4 | 5 | N |
| 1 | 2 | 3 | 4 | 5 | 2.Ability to work effectively with people of different cultures / backgrounds 能與不同文化或背景的人共事 | 1 | 2 | 3 | 4 | 5 | N |

## II. Curriculum and Campus Life 課程及校園生活

Major What was your major during your study in the University? (Please tick the most appropriate box) 在求學時，您主修哪個學科？（請在適當空格內填上"✓"號）

1.Art 文科:
- ☐ 1.Chinese 中文
- ☐ 2.Contemporary English Studies 當代英語語言文學
- ☐ 3.Contemporary English and Education 當代英語語言文學與教育學
- ☐ 4.Cultural Studies 文化研究
- ☐ 5.History 歷史
- ☐ 6.Philosophy 哲學
- ☐ 7.Translation 翻譯
- ☐ 8.Visual Studies 視覺研究

2.Business Administration 工商管理:
- ☐ 9.Accounting 會計
- ☐ 10.Finance 財務
- ☐ 11.General Business Management 企業管理
- ☐ 12.Human Resource Management 人力資源管理
- ☐ 13.Logistics and Decision Science 物流與決策科學
- ☐ 14.Information Systems 資訊系統
- ☐ 15.Marketing 市場學
- ☐ 16.Risk and Insurance Management 風險及保險管理

3.Social Sciences 社會科學:
- ☐ 25.Economics 經濟學
- ☐ 26. Political Science 政治學
- ☐ 27. Psychology 心理學
- ☐ 28. Sociology 社會學
- ☐ 29. China and Asia Pacific Studies 中國與亞太研究
- ☐ 30. Social and Public Policy Studies 社會與公共政策研究
- ☐ 17. Behavioural Science in Modern Society 現代行為科學 (BSMS)
- ☐ 18. China and Asian Pacific Affairs 中國與亞太關係(CAPA)
- ☐ 19. Contemporary Economic and Public Policy 當代經濟與公共政策 (CEPP)
- ☐ 20.Contemporary Social Issues and Policy 當代社會問題與政策研究 (CSIP)
- ☐ 21.International Political and Economic Affairs 國際政治經濟事務(IPEA)
- ☐ 22.International Studies 國際研究 (IS)

Others (Please Specify) 其他（請註明主修科）: _______________________

| P2.Please rate your learning experience regarding your programme at the University. Please indicate your response to each question by circling the appropriate score. 請評價您在大學修讀有關學科的經驗：（請為以下各問題圈出適當的分數。） | Strongly Disagree → Strongly Agree 極不同意 → 極同意 | | | | | Unable to rate 不適用 99 |
|---|---|---|---|---|---|---|
| 1.I found attending classes useful and productive. 您認為課堂有用和富有成效 | 1 | 2 | 3 | 4 | 5 | N |
| 2.I had a clear sense on the courses' pace and expectations. 您清楚知道課堂的進度和要求 | 1 | 2 | 3 | 4 | 5 | N |
| 3.The teaching staff motivated me to try my best. 老師激勵您做到最好 | 1 | 2 | 3 | 4 | 5 | N |
| 4.The teaching staff were very understanding about the difficulties I might encounter with my work. 老師能體恤同學學習中遇到的難題 | 1 | 2 | 3 | 4 | 5 | N |
| 5.The teaching staff usually gave me helpful feedback. 老師常常給予您有用的回應及答覆 | 1 | 2 | 3 | 4 | 5 | N |
| 6.The teaching staff worked hard to make their subjects interesting. 老師盡力使教學變得有趣 | 1 | 2 | 3 | 4 | 5 | N |
| 7.The teaching staff clearly specified their expectations at the beginning of the course. 老師對同學的要求一開始就交待清楚 | 1 | 2 | 3 | 4 | 5 | N |
| 8.The programme developed my problem-solving skills. 課程提高您解決問題的能力 | 1 | 2 | 3 | 4 | 5 | N |
| 9.The courses I took sharpened my analytical skills. 課程加強您的分析能力 | 1 | 2 | 3 | 4 | 5 | N |
| 10.The courses I took helped me develop my ability to work as a team member. 課程幫助您學會和其他人建立團隊精神 | 1 | 2 | 3 | 4 | 5 | N |
| 11.The courses I took helped me to develop my planning ability. 課程提高您的規劃能力 | 1 | 2 | 3 | 4 | 5 | N |
| 12.The courses I took enhanced my employment opportunities. 課程有助增加您的就業機會 | 1 | 2 | 3 | 4 | 5 | N |
| 13.English for Communication I & II strengthened my fluency in written English. 英文傳意課程能令您增強英語書寫能力 | 1 | 2 | 3 | 4 | 5 | N |
| 14.English for Communication I & II strengthened my fluency in Oral English. 英文傳意課程能令您增強英語會話能力 | 1 | 2 | 3 | 4 | 5 | N |
| 15.English for Communication I & II made me feel confident to use English effectively in my work. 英文傳意課程令您能夠在工作中自信和有效地運用英語 | 1 | 2 | 3 | 4 | 5 | N |
| 16.Practical Chinese I & II strengthened my ability to write formal Chinese documents. 實用中文課程增強您的實用文寫作能力 | 1 | 2 | 3 | 4 | 5 | N |
| 17.Practical Chinese I & II helped me to communicate effectively using Putonghua. 實用中文課程增強您的普通話說話應對能力 | 1 | 2 | 3 | 4 | 5 | N |
| 18.The General Education / Core Curriculum courses were very useful. 通識教育課程／核心課程非常有用 | 1 | 2 | 3 | 4 | 5 | N |
| 29.The General Education / Core Curriculum helped in your work life 通識教育課程／核心課程對您的工作有幫助 | 1 | 2 | 3 | 4 | 5 | N |

| P2 **Please rate your time. Please indicate your response to each question by circling the appropriate score.** 請評價您在大學生活的經驗：（請為以下各問題圈出適當的分數。） | Strongly Disagree → Strongly Agree 極不同意 → 極同意 | | | | | Unable to rate 不適用 99 |
|---|---|---|---|---|---|---|
| 19.The Integrated Learning Programme helped my whole-person development (e.g. Civil Education, Intellectual Development, Physical Education, Social and Emotional Development, and Aesthetic Development). 綜合學習課程有助您的全人發展 (德、智、體、群、美等) | 1 | 2 | 3 | 4 | 5 | N |
| 20.Participating in Student Organizations developed my leadership skills. 參與學生組織有助提升您的領袖才能 | 1 | 2 | 3 | 4 | 5 | N |
| 21.Community Services/ Service-Learning gave me a chance to contribute to society. 參與社區服務/ 服務研習計劃有助您將所學到的知識與技能回饋社會 | 1 | 2 | 3 | 4 | 5 | N |
| 22.International Exchange Programmes broadened my international perspectives. 國際交流計劃擴闊了您的國際視野 | 1 | 2 | 3 | 4 | 5 | N |
| 23.Hostel life strengthened my interpersonal skills. 舍堂生活增強您的人際技巧 | 1 | 2 | 3 | 4 | 5 | N |
| 24.Hostel life made me more caring and responsive to the well-being of others. 舍堂生活使您學會處處關懷別人 | 1 | 2 | 3 | 4 | 5 | N |
| 25.Hostel life helped my whole-person development. 舍堂生活使您的全人發展更完滿 | 1 | 2 | 3 | 4 | 5 | N |

| P2 **Relationship between your learning experience and your engagement at work.** Please indicate your response to each question by circling the appropriate score. 您在大學的經驗和您投入學業的關係：（請為以下各問題圈出適當的分數。） | Strongly Disagree → Strongly Agree 極不同意 → 極同意 | | | | | Unable to rate 不適用 99 |
|---|---|---|---|---|---|---|
| 27.I had at least one professor at the campus who cared about me as a person, made me excited about my learning and encouraged me to pursue my dreams. 在大學學習期間，最少有一位教授會關心我，令我投入學習和鼓勵我追求夢想 | 1 | 2 | 3 | 4 | 5 | N |
| 28.I am highly engaged and motivated at work 我非常投入及熱愛我的學業 | 1 | 2 | 3 | 4 | 5 | N |

P2_29.Overall, how would you rate your learning experience? (Please tick the most appropriate box) 整體來證，您如何評價在大學的學習生活？（請圈出適當的分數）

| Very Disappointing 極失望 | Disappointing 失望 | Neutral 一般 | Rewarding 有意義 | Very Rewarding 極有意義 | P2_Other. Any |
|---|---|---|---|---|---|
| 1 | 2 | 3 | 4 | 5 | |

Other Comments 其他意見:

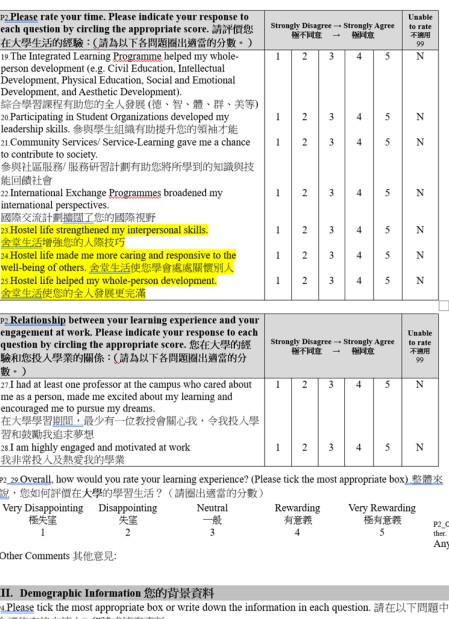

### III. Demographic Information 您的背景資料

P4.Please tick the most appropriate box or write down the information in each question. 請在以下問題中最合適的空格內填上"✓"號或填寫資料。

1. Gender 性別: □ 1 Male 男 □ 2 Female 女

2. Year of Graduation 畢業年份: □ 2013 □ 2014 □ 2015 □ 2016 □ 2017

3. Age 年齡: □ 1.22-23 □ 2.24-25 □ 3.26-35 □ 4.36 or above36 歲或以上

4. 1.Have you taken additional programmes after your graduation? 您有否在畢業後修畢其他課程而獲得學歷？
　　□ 1.No 沒有
　　□ 2.Yes 有 (c1.Please select the following 請選出以下學歷)
　　　　□ 1.Bachelors 學士　　　□ 2.Masters 碩士　　□ 3.Doctorate 博士
　　　　□ 4.Certificate 證書　　　□ 5.Diploma 文憑　　□ 6.Postgraduate Diploma 深造文憑
　　　　□ 7.Others 其他 (Please Specify 請列明): _______________

5. 1.Have you been awarded further qualification after your graduation? 您有否在畢業後修畢其他課程而獲得專業資格？
　　□　　1.No 沒有
　　□　　2.Yes 有 (Please select the following 請選出以下資格)
　　　　□ 1.ACCA/HKICPA　　□ 2.CFA　　　□ 3.CFP　　□ 4.FRM
　　　　□ 5.HKICS　　　□ 6.HKSI　　　□ 7.IIQE　　□ 8.ANZIIF
　　　　□ 9.LOMA　　　　□ 10.Registered Social Worker 註冊社工
　　　　□ 11.Education Diploma 教育文憑　　□ 12.Law 法律學位
　　　　□ 13.Property Agency Licence 地產代理牌照
　　　　□ 14.Professional IT Certification 專業 IT 証書(e.g. MCSE, MCSA, CCNP, CCNA)
　　　　□ 15.Others 其他(Please Specify 請列明): _______________

### IV. Job and Career information 工作及職業資料

P5.Please tick the most appropriate box or write down the information in each question. 請在以下問題中最合適的空格內填上"✓"號或填寫資料。

1. How many full-time jobs have you taken since you graduated? 自大學畢業後，您做過多少份全職工作？

2. Current Employment Status 現時的就業情況:
　　□ 1.Full Time 全職　　□ 2.Part Time 兼職　□ 3.Self-Employed 自僱　□ 4.Owner 東主
　　□ 5.Further Studies 全　□ 6.Unemployed 待　□ 8.Working Holiday 工
　　職進修　　　　業　　　　作假期
　　□ 7.Others 其他 (Please Specify 請列明): _______________

3. (If the answer in Q2 is Full Time / Part Time / Self-Employed / Owner) What is your industry and position? (如 Q2 回答全職/兼職/自僱/東主) 請問您服務的機構所屬行業及職位是什麼呢？
　　Industry 行業: _______________
　　Position 職位: _______________

4. Your Current Salary 您的現時月薪
　　[(Basic Salary + Commission + Allowance) X Times of Payment per year / 12]
　　[(基本月薪 + 佣金 + 津貼) x 每年糧期次數 / 12 ]:
　　□ 1.HK$ 5,000 below 以下　　　□ 2.HK$ 5,000 - HK$ 10,000
　　□ 3.HK$ 10,001 - HK$ 15,000　　□ 4.HK$ 15,001 - HK$ 20,000
　　□ 5.HK$ 20,001 - HK$ 25,000　　□ 6.HK$ 25,001 - HK$ 30,000
　　□ 7.HK$ 30,001 - HK$ 35,000　　□ 8.HK$ 35,001 - HK$ 40,000
　　□ 9.HK$ 40,001 or above 以上

| 5. Size of company you work for 您服務的機構聘用的全職僱員數目 | Number of Employees 僱員數目: |
|---|---|
| 1. If your employer is a multinational enterprise, please tick 如服務的機構為 **跨國企業**，請在方格填上"✓"號 (Number of employees based in HK 計算僱員的數目，只限於香　港分公司) □ | 1.Less than 20 少於 20　□ |
| 2. If your employer is a local enterprise, please tick 如服務的機構為 **本地企業**，請在方格填上"✓"號 (Number of employees based in HK, Macau, Taiwan, Mainland, and other overseas offices 計算僱員的數目，包括港、澳、台、大陸及海外各分公司) □ | 2.21-50　　□ 3.51-100　　□ 4.101-200　　□ 5.201-500　　□ 6.More than 500 多於 500　□ |
| 3. If your employer is a Chinese enterprise, please tick 如服務的機構為 **中資企業**，請在方格填上"✓"號 (Number of employees based in HK 計算僱員的數目，只限於香港分公司) □ | 7.Not sure 不清楚　　□ |

6. Have you had any job promotions during your full time employment since you graduated? 自大學畢業後，在您全職工作期間，有沒有曾經晉升呢？
　　□ No 沒有
　　　□ Yes 有, (number of promotions 多少次呢？) _______

7. What is/are the reason(s) that might lead you to change your job in the future? 未來會令您轉工 的原因會是什麼呢？(you may tick more than one box if appropriate 可選取多於一項)
   ☐ 1.Career growth 職業改進     ☐ 2.Promotion opportunities 晉升機會
   ☐ 3.Salary 薪酬     ☐ 4.Workload/work pressure 工作量 / 工作壓力
   ☐ 5.Management support 管理層支持   ☐ 6.Family issue 家庭因素
   ☐ 7.Looking for challenges 尋求挑戰
   ☐ 8.Enhanced professional qualification 考取專業資格
   ☐ 9.Others 其他 (Please Specify 請註明): ___________________

8. What is your career plan in the coming year? 在未來數年，您的職場計劃是怎樣？
   ☐ 1. Stay on the current job 留在現有工作 ☐ 2.Seek for a job change 轉工
   ☐ 3. Working holiday 工作假期     ☐ 4. Further studies 持續進修
   ☐ 5.Others 其他 (Please Specify 請註明): ___________________

**The following information will be used for Survey of Employers**
**以下資料會用作大學僱主問卷調查之用**

9a. Current Employer 現任僱主機構資料:
   Name 名稱: ___________________
   Address 地址: ___________________
   Tel 電話號碼: _____________ Fax 傳真號碼: _____________

9b. Former Employer:
   Name 名稱: ___________________
   Address 地址: ___________________
   Tel 電話號碼: _____________ Fax 傳真號碼: _____________

Other: Any Other Comments 其他意見:
___________________

**As a token of appreciation for your support, if you finish this questionnaire, you will be eligible to enter into the Lucky Draw to win an IPad. If you want to participate in the Lucky Draw, please fill in the *personal information below (optional):** 為答謝校友的參與，完成問卷調查的校友可參加抽獎，贏取 iPad 乙部。如欲參加抽獎，請填寫以下*個人資料（選擇性填寫）:

**Name 姓名:** ___________________

**Telephone number 電話號碼:** ___________________

**Email address 電郵地址:** ___________________

~**End** 問卷完~

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
