# Peer review of "A Novel Influence Analysis-Based University Major Similarity Study"

_education, doi:10.3390/educsci14030337_

Round 1

Reviewer 1 Report

Comments and Suggestions for Authors

I report an original methodology and approach to data analysis, which, however, can be somewhat confusing if its relevance and why it was chosen to be used is not thoroughly explained. The objective of the study is clear, but one is left with the feeling, reading the paper, that the aims are more ambitious and were not clearly explained. 

In the same way, I suggest that a more exhaustive discussion could be written, as well as a description of the limitations of the study in the conclusion. 

Comments on the Quality of English Language

In general, I believe that the language issue is clear and there is no major problem in this area. 

Reviewer 2 Report

Comments and Suggestions for Authors

It’s a great effort of the authors to address an important issue of presents the study on Influence Analysis in the context of university majors. However, there are several aspects that could be considered:

Title: the term “Influence Analysis” seems to be the main topic of the manuscript; however, it appears only three times (in the title, abstract and keywords). This term does not appear throughout the whole manuscript even once. Please consider the consistent use of terms. The terms “influence matrix” seems used more frequently over the manuscript.

Abstract: it could benefit from providing more specific details about the methodology and findings. Including specific findings from the empirical study would enhance the reader's understanding of the research. The abstract effectively highlights the relevance of the research topic in the era of big data and emphasizes the potential insights that can be gained from studying university major similarities. However, it could further elaborate on the significance of the findings and how they contribute to existing literature or address practical implications in higher education.

1.       INTRODUCTION

Lines 27-32: please add some references to support your arguments.

Lines 58-66: please add some references to support your arguments.

Lines 86-92: please consider moving this paragraph to the MATERIAL AND METHOD section.

Lines 93-96: please consider removing it.

2.       OVERVIEW OF DATA MODEL

Lines 117-133: please consider adding some references to support your arguments.

Line 206: please add some references to support your arguments.

Lines 107-109: it is no clear criteria for “stubborn nodes” or “leader majors”.

In this section, it not very clear which part of the Opinion Dynamic Model presented is the novel of this study, differentiated from the previous DeGroot model and the authors’ previous work [11]. The captions of the Fig. 1-3 need to be specified if they are newly proposed images by the authors, or newly drawn by the authors based on existing knowledge.

3.       MATERIALS AND METHODS

Please clarify if the authors used the secondary data provided by the university, publicly available data, or primary data collected by the authors per se.

Line 242:  About Demographic information, how the confidentiality of private information could be achieved. Please justify why the IRB approval is not applicable (line 390).

Line 244: it said “over 30 majors”, however, the analysis was done on 29 majors (line276, 289, 290). In the Appendix A (pages 13,14), there were 30 majors listed, however, the majors 23 and Major 24 were lacked. Were the number and names of majors changed over the period 2002-2020? Please make sure about consistency and add the missing majors into the Appendix.

Line 260: How many valid responses per question were included into analysis?

Lines 289-290: lack of criteria for selection of a “leader major”. It needs to justify the threshold for choosing 22/7.

4.       RESULTS

Lines 325, 352, 355, 358: please renumbering the figures, It should be Fig 4,5,6,7.

Please specify the method to produce these figures (except the last figure, which was produced by Gephi), and add this method to the Material and Methods section.

Lines 360: please add references for the algorithm and/or for the software.

5.       CONCLUSION:

Based on the information provided, the main findings appear to be focused on the application of the opinion dynamic model to characterize and infer the influence relationships among university majors based on questionnaire data. While the manuscript mentions the conduct of an empirical study involving the collection of questionnaire data from graduates across different majors, it does not specify specific results from this empirical analysis. Instead, it seems emphasized the use of the opinion dynamic model as the primary methodology to examine university major similarities and infer influence relationships among them.

Lines 376-381: the findings presented as “Furthermore…. Major” seem not supported by the results presented in the 4. RESULTS section.

Language: While the manuscript is generally well-written, there are instances where the language could be further refined for clarity, ensuring consistency in terminology. The abstract could be more concise by trimming down redundant phrases and unnecessary details. E.g., the phrase "Given the intricate nature of interpersonal interactions" could be simplified to "Due to the complexity of social interactions."

Author Response

Please find the attached doc.

Round 2

Reviewer 1 Report

Comments and Suggestions for Authors

I appreciate that you have taken the comments into account and without a doubt, the abstract and conclusions were improved. It is also now clearer to understand the instrument you used.